# Rheumatoid Arthritis and Osteoporosis as Prototypes of Immunosenescence in Osteoimmunology: Molecular Pathways of Inflammaging and Targeted Therapies

**DOI:** 10.3390/ijms26199268

**Published:** 2025-09-23

**Authors:** Ernesto Aitella, Gianluca Azzellino, Ciro Romano, Lia Ginaldi, Massimo De Martinis

**Affiliations:** 1Department of Life, Health and Environmental Sciences, University of L’Aquila, 67100 L’Aquila, Italy; gianluca.azzellino@aslteramo.it (G.A.); massimomariamarcello.demartinis@univaq.it (M.D.M.); 2Clinical Immunology Outpatient Clinic, Division of Internal Medicine, Department of Advanced Medical and Surgical Sciences, “Luigi Vanvitelli” University of Campania, 80131 Naples, Italy; ciro.romano@unicampania.it

**Keywords:** immunosenescence, inflammaging, osteoimmunology, rheumatoid arthritis, senile osteoporosis, gerontorheumatology, elderly, biological therapies

## Abstract

Immunosenescence refers to the set of immunoendocrinological alterations underlying the progressive decline in innate and adaptive immune function that occurs with aging. It is closely linked to the concept of inflammaging, a state of low-grade chronic systemic inflammation that contributes to age-related diseases. In the elderly, key features of diseases such as rheumatoid arthritis, particularly in its elderly onset form, and senile osteoporosis are characterized by a decline in sex hormones and the immunoregulatory IL-2; an increase in serum autoantibodies and pro-inflammatory mediators such as TNF-α, IL-6; and upregulation of bone-related factors RANKL, DKK1, and sclerostin, including the dysregulation of the IL-33/IL-31 axis. The aim of this review is to examine the key molecular pathways of immunosenescence in osteoimmunology, as well as the potential for therapeutic modulation of inflammaging through biologic and target synthetic disease-modifying antirheumatic drugs, denosumab and romosozumab, with particular attention to their management in elderly patients.

## 1. Introduction

Aging is characterized by immunoendocrinological alterations that significantly impact morbidity and mortality [1,2]. The primary hormonal and immunological changes lead to substantial modifications in immune cell function, including serum concentrations of antibodies and changes in the cytokine environment [3]. Immunosenescence results in a progressive decline in immunocompetence and tumor surveillance, as well as an increased susceptibility to dysregulated immune responses, autoimmune conditions, metabolic syndromes, and chronic inflammatory diseases [4].

Simultaneously, factors such as hyperparathyroidism, decreased levels of vitamin D and reduced osteoblast activity contribute to an elevated risk of osteoporosis [5,6]. Furthermore, the interplay between metabolic pathways, genetic, and epigenetic factors, and lifelong environmental exposures under stress conditions, gives rise to the emerging concept of “inflammaging,” which plays a central role in the development of age-related diseases [7].

For the elaboration of this manuscript, a literature search was conducted using the PUBMED database, selecting original articles and reviews in English—preferentially in the last ten years—with the keywords rheumatoid arthritis, osteoporosis, immunosenescence, inflammaging, elderly, biological therapies, and target synthetic disease-modifying antirheumatic drugs, focusing on pathophysiology, molecular aspects, clinical features, and therapeutic strategies in aging.

The aim of this review is to examine the key molecular pathways of immunosenescence, highlighting the convergence of the physiopathological processes and their major immunological and rheumatological correlates, particularly in the context of osteoimmunology. This analysis is approached from a translational standpoint, focusing on the clinical and therapeutic implications for elderly patients. Rheumatoid arthritis (RA) and osteoporosis are analyzed as prototypical examples of immuno-skeletal interface diseases, with particular attention to their molecular mechanisms and interconnections, and to the role of targeted and advanced therapies in modulating inflammaging in the immunosenescent population [8].

## 2. Immunosenescence and Inflammaging: The Contribution to Rheumatic Disorders

The aging process can be conceptualized as a longitudinal vicious spiral linking birth and death through the dynamic interplay between the endocrine and immune systems [9], while the resulting set of processes defines immunosenescence at cellular level [10,11]. Age-related hormonal changes affect the pituitary, pineal, and adrenal glands, as well as the gonads and parathyroid glands. Senescence is typically characterized by decreased levels of melatonin, growth hormone (GH), insulin-like growth factor 1 (IGF-1), dehydroepiandrosterone (DHEA), testosterone, 17β-estradiol, and vitamin D3. In contrast, serum concentrations of cortisol and parathyroid hormone increase with age [12,13,14,15].

From a hormonal perspective, adulthood begins with a scenario typically characterized by a testosterone/estrogen dualism, with differing effects on immune function [16]. Testosterone primarily exerts immunosuppressive effects, while estrogens and progesterone tend to enhance immune responses. This dichotomy contributes to sex-based differences in immune function. In males, testosterone dominance is associated with a Th1-skewed immune profile [17] and characterized by increased CD8^+^ T cell cytotoxicity and enhanced immunosuppressive activity of neutrophils, and decreased natural killer (NK) cell activity, B cell maturation, and antibody production. Moreover, testosterone is linked to elevated IgG4 levels and reduced expression of major histocompatibility complex (MHC) molecules. In contrast, estrogens promote a Th2-dominant immune response by enhancing B cell survival, maturation, isotype switching, and antibody production; they also reduce the negative selection of high-affinity autoreactive lymphocytes, enhance dendritic cell activity, and increase CCR5-mediated T cell homing: these are the mechanisms that heighten inflammatory responses and contribute to the pathogenesis of autoimmune diseases such as RA, systemic lupus erythematosus, and multiple sclerosis [18].

With aging, the regulatory influence of sex hormones on immune function shifts, resulting in a remodeled immunological landscape. Among the most relevant hormones, DHEA is the most abundant circulating steroid, synthesized mainly in the adrenal glands, and serve as a precursor to testosterone, estrone, and estradiol. Age-related declines in DHEA contribute to both immunological and systemic changes. Cardiovascular effects include increased low-density lipoprotein (LDL) levels and heightened prothrombotic vascular activity. In the skeletal system, DHEA deficiency reduces osteoblast longevity and prolongs osteoclast survival, increasing osteoporosis risk. Immunologically, reduced monocyte activity occurs alongside paradoxically elevated antibody production [19,20].

In particular, two major cytokine shifts associated with aging have critical relevance. The first is a decline in interleukin-2 (IL-2), which impairs T cell development and limits proliferative capacity, both of which are key features of immunosenescence. The second is an elevation in interleukin-6 (IL-6), a pro-inflammatory cytokine implicated in chronic inflammation and the pathogenesis of age-related diseases such as atherosclerosis, RA, SLE, inflammatory bowel disease, and osteoporosis [21,22].

Interestingly, IL-2 and IL-6 are characterized by bidirectional effects on immune regulation, with concentration-dependent outcomes that may even be opposite, for example, in immune activation versus immune tolerance or in bone destruction versus bone repair, respectively [23,24,25,26]. In this manuscript, we specifically refer to the decrease in IL-2 and the increase in IL-6 in the context of pathological aging in RA and osteoporosis, as validated in animal models [27,28].

Aging also disrupts the hypothalamic–pituitary–adrenal (HPA) axis, altering both the quantity and circadian rhythm of basal cortisol secretion. Older adults often exhibit elevated evening cortisol levels and a flattened daily rhythm [29]. Chronic hypercortisolemia has long-term immunological, neurological, and metabolic consequences. It promotes a Th2-biased immune profile with elevated levels of IL-3, IL-4, IL-5, IL-6, and IL-10, as well as increased antibodies production. Neurologically, it contributes to cognitive decline, hippocampal damage, and a higher risk of dementia. Metabolic effects include increased adiposity, loss of lean body mass, reduced bone mineral density, and progression of atherosclerosis, partly due to diminished protective effects on lipid metabolism and vascular integrity. Additionally, chronic cortisol excess impairs glucose metabolism, leading to insulin resistance and reduced glucose tolerance [30].

Collectively, these endocrine alterations drive a progressive decline in immunocompetence, compounded by reduced anabolic hormone levels and persistent inflammatory and neuroendocrine dysregulation. The immunological landscape of aging is profoundly shaped by these hormonal shifts. The decline in anabolic hormones like DHEA and testosterone, combined with elevated cortisol and pro-inflammatory cytokine expression, promotes a progressive decline in immune competence. This immunodeficiency increases vulnerability to infections, chronic inflammation, autoimmune diseases, and neurocognitive decline [10,31].

Another more recent concept is represented by inflammaging, in which age-related immune changes are reorganized and reconceptualized in response to lifelong stress. In this context, pro-inflammatory processes accumulate across the lifespan, potentially leading to adaptive or damaging outcomes, that can take either a beneficial or detrimental direction. This dichotomous model suggests that, depending on the dominant pathway, individuals may either support mechanisms that promote longevity or, conversely, develop age-related diseases, such as Alzheimer’s disease, osteoporosis, diabetes, and atherosclerosis [32]. Inflammaging refers specifically to age-associated, systemic, low-grade inflammation, sustained by immunosenescence and shaped by lifelong environmental exposures, genetic predisposition, and epigenetic regulation. On one side of the model, there are the “stressors”, such as infections, metabolic stress, oxidative damage, chemical agents, and physical injuries. On the other side, there are molecular mechanisms that determine whether aging progresses in a healthy or pathological direction. As in the immunological paradigm based on the two-signal theory, inflammaging may involve a “second-hit” mechanism: beyond external stressors—first hit—the presence or absence of protective robust or fragile genetic variants—second hit—becomes critical in determining whether aging mechanisms are mitigated or exacerbated [33].

Interestingly, this chronic inflammatory state of inflammaging is a sterile inflammation, in which the aforementioned self and non-self-stimuli act as damage-associated molecular patterns (DAMPs) and pathogen-associated molecular patterns (PAMPs), respectively, and interact with receptors such as Toll-like receptors (TLRs) and NOD-like receptors, activating the innate immune system [11]. In this context, the microbiota also seems to play a role [34], inducing a form of low-grade sterile chronic inflammation known as metaflammation, which underlies the metabolic inflammation observed in metabolic diseases [35]. A key cell of the innate immune system in the context of inflammaging, however, is thought to be the macrophage, whose stress response determines its activation and the progressive increase in inflammation with age, typical of the so-called “macroph-aging” [36].

In the context of the cellular and molecular aspects of aging, another significant hallmark of immunosenescence is mitochondrial dysfunction. In this setting, DAMPs released during cellular damage, necrosis, or apoptosis, such as mitochondrial reactive oxygen species (mtROS) and mitochondrial DNA (mtDNA), are recognized by TLRs. Plasma levels of mtDNA gradually increase after the age of 50, accounting for the pro-inflammatory contribution of this pathway during the aging process. At the same time, degeneration of the mitochondrial antiviral signaling protein (MAVS), anchored to the mitochondrial outer membrane, may explain the reduced synthesis of type I interferons and the increased susceptibility of elderly individuals to viral infections [37].

At the basis of these cellular alterations of immunosenescence lies the role of hematopoietic stem cells (HSCs), which serve as the reservoir for immune cell replenishment throughout life. The dysfunction of HSCs, up to their exhaustion, in fact, leads to a reduced capacity for self-renewal of their pool and an alteration of their heterogeneity, with a preferential decline in lymphoid-biased cells as compared to those of the myeloid lineage. This imbalance has downstream consequences on T and B lymphocyte differentiation and, ultimately, on immune function [38].

The interplay between traditional immunosenescence frameworks and the re-elaboration offered by the inflammaging model is illustrated in Figure 1.

Immunosenescence is depicted as a spiral connecting birth and death, encompassing the age-related changes associated with inflammaging. The upper and lower triangular boxes illustrate the main immunoendocrine and cellular alterations associated with immunosenescence, categorized, respectively, as increases and decreases, along with the two major cytokine changes in pathological aging: the reduction in IL-2 and the increase in IL-6. Based on genetic predisposition, epigenetic regulation, and molecular and metabolic pathways, lifelong environmental interactions and exposure to antigenic, oxidative, metabolic, chemical, and physical stressors will determine the trajectory toward either longevity or pathological aging, in accordance with the “second hit” model.

PTH: parathyroid hormone; LDL: low-density lipoprotein; GH: growth hormone; IGF-1: insulin-like growth factor 1; DHEA: dehydroepiandrosterone; HSCs: hematopoietic stem cells. 

## 3. Rheumatoid Arthritis in the Elderly: From Molecular and Cellular Basis to Clinical Features

RA is a systemic autoimmune disease with a predominantly erosive nature, primarily affecting diarthrodial joints through the involvement of the synovial tissue and synoviocytes. Approximately one-third of RA cases have their onset in older adults, with a rising incidence between the fifth and sixth decades of life, a peak prevalence around the age of 80 [39], and an increasing trend [40]. In elderly onset RA (EORA), the traditional gender disparity tends to diminish, occasionally showing a slight male predominance [41].

The elderly RA population includes both individuals with long-standing disease that began earlier in life (young-onset RA, YORA) and those with new-onset RA after the age of 60 [42]. This group frequently presents with multiple comorbidities—particularly cardiovascular diseases, hepatic or renal dysfunction, cognitive impairment, polypharmacy, and increased susceptibility to infections and malignancies—which complicate therapeutic strategies and contribute to overall frailty. Moreover, EORA is often associated with a poorer prognosis, partly due to more conservative treatment approaches driven by safety and tolerability concerns [43].

In RA, age-related immune alterations result in a relative decline in adaptive immune function with a compensatory increase in non-specific innate responses. Peripheral T cells, in response to diminished thymic output, undergo repeated antigen-driven proliferation, leading to senescence and functional exhaustion. These senescent T cells exhibit a restricted T-cell receptor (TCR) repertoire, loss of CD28 expression, and impaired immune tolerance, contributing to increased susceptibility to autoimmunity, as well as RA [44]. Senescent T cells in RA also show premature telomere shortening, cell cycle arrest, diminished clonal expansion, and altered effector functions. Deficient DNA repair mechanisms lead to the accumulation of genomic damage, apoptosis, and lymphopenia. This immunological dysfunction is further characterized by the expansion of dysfunctional regulatory T cells and pro-inflammatory effector T cell subsets [45]. Importantly, these immune alterations appear to be primary features of immunosenescence rather than merely consequences of chronic inflammation. HLA-DRB1 genotype distributions differ between YORA and EORA: younger-onset RA is often associated with HLA-DRB1*01 and *04 alleles, while HLA-DRB113 and *14 are more prevalent in elderly patients, particularly with seronegative RA or polymyalgia rheumatica-like symptoms [46,47]. HLA-B27 positivity has also been observed in cases of remitting seronegative symmetrical synovitis with pitting edema (RS3PE), although HLA-B27 is classically associated with ankylosing spondylitis [48].

RA patients display accelerated immunosenescence compared to age-matched healthy individuals, with evidence of a thymic involution rate equivalent to that of individuals 20 years older, alongside reduced T cell diversity and cellular aging [44,49]. Compared to YORA, EORA patients typically have higher serum IL-6 levels, lower TNF-α levels, reduced rheumatoid factor titers, elevated acute-phase reactants, and increased anti-cyclic citrullinated peptide (anti-CCP) antibody titers [39]. Clinically, the disease often presents acutely and may mimic a parainfectious process, which can delay diagnosis and initiation of treatment. These patients also tend to exhibit more systemic features. Due to elevated inflammatory markers in the elderly, clinical activity scores such as disease activity score based on 28 joints (DAS28) must be interpreted cautiously, as they may underestimate true remission. In this context, subclinical inflammation markers such as the platelet-to-lymphocyte ratio (PLR) and neutrophil-to-lymphocyte ratio (NLR) can assist in evaluating disease activity. However, aging is associated with decreased platelet counts despite increased platelet activation, which can result in relatively low PLR values. NLR levels in EORA are generally similar to those in YORA and tend to rise during active disease phases, though their utility in defining remission is limited by low sensitivity [50].

Based on these molecular and cellular characteristics, EORA can be clinically classified into three principal phenotypes. The classic form resembles typical RA with predominant involvement of peripheral joints. A second phenotype mimics polymyalgia rheumatica, with predominant involvement of proximal joints [51]. The third and distinct presentation is RS3PE, characterized by acute, symmetric synovitis and tenosynovitis with pitting edema in the hands and feet. This phenotype is notably seronegative for both RF and anti-CCP antibodies and typically does not lead to joint erosions. RS3PE generally responds well to low-dose corticosteroid therapy, with remission occurring within three to eighteen months [48].

### Conventional and Advanced Therapies in Rheumatoid Arthritis: Considerations for the Elderly

Currently, geriatric population is characterized by longer life expectancy, multiple comorbidities, and widespread polypharmacy. However, there is a significant lack of real-world data on the management of rheumatic diseases in this age group, particularly regarding the use and impact of conventional and innovative disease-modifying therapies—both in terms of drug interactions and immunological impact on elderly [52].

A recent single-center study from Turkey analyzed the epidemiological and clinical features of a large elderly patient cohort with rheumatic diseases, providing a real-world scenario [53]. In this study, rheumatic diseases were identified in over half of the elderly patients, who were predominantly female, with a significant representation of elderly onset. The most frequent diagnoses were RA and polymyalgia rheumatica, followed by connective tissue diseases, particularly Sjögren’s syndrome. Comorbidities were common and included cardiovascular disease, diabetes mellitus, chronic kidney disease, thyroid disorders, and respiratory conditions such as allergic asthma, and cancer. Regarding treatment, in this casuistry, the most frequently prescribed medications were corticosteroids, followed by non-steroidal anti-inflammatory drugs (NSAIDs) and conventional synthetic disease-modifying antirheumatic drugs (csDMARDs) such as methotrexate, leflunomide, sulfasalazine, and hydroxychloroquine. These molecules act respectively by inhibiting dihydrofolate reductase; inhibiting dihydroorotate dehydrogenase; and exerting anti-inflammatory and immunosuppressive effects through the metabolites sulfapyridine and 5-aminosalicylic acid and through lysosomal alkalinization and immunomodulatory interference with antigen processing [54]. On the other hand, the use of anti-tumor necrosis factor (TNF) agents and other biologic DMARDs (bDMARDs) was relatively low, while targeted synthetic DMARDs (tsDMARDs) like tofacitinib and baricitinib were used only in a few cases.

Therefore, elderly patients with RA are often prescribed corticosteroids more frequently; however, both glucocorticoids and NSAIDs are not recommended as first-line treatments due to their significant adverse effects, including gastrointestinal toxicity and increased cardiovascular risk. Chronic corticosteroid use is associated with increased all-cause mortality and fracture risk [55], leading international guidelines to advocate for steroid-sparing strategies, especially during maintenance phases. Methotrexate remains the cornerstone csDMARD in EORA, and is often used even more frequently than in YORA, albeit at lower doses [56]. Among csDMARDs, hydroxychloroquine is commonly prescribed and leflunomide is generally well-tolerated, while sulfasalazine is used less frequently in this age group [57].

Although combination or advanced therapies are less commonly employed in elderly patients, biologic monotherapy becomes a valuable alternative in cases of intolerance or insufficient response to csDMARDs. Biologic agents have demonstrated comparable efficacy in both younger and older populations; however, anti-TNF agents are more frequently discontinued in the elderly due to adverse events rather than lack of effectiveness. In some cases, functional outcomes lag behind clinical improvement, likely due to coexisting osteoarthritis or other age-related conditions [58].

From a pathophysiological perspective, IL-6 inhibition by Tocilizumab appears theoretically appropriate in EORA, given the central role of IL-6 in RA pathogenesis and its elevated levels in elderly patients. However, a higher risk of infections [59] have been observed in this population. Similar concerns apply to rituximab, which induces B cell depletion through binding to the B cell surface protein CD 20 [60,61].

In contrast, Abatacept has shown a favorable safety and tolerability profile in older adults, with lower discontinuation rates due to adverse events and a reduced risk of infections compared to infliximab or etanercept [62,63]. Moreover, the pharmacological modulation of the CTLA-4 pathway by Abatacept, binding to CD80 and CD86 on antigen-presenting cells (APCs), may be particularly beneficial in EORA due to the enhanced humoral autoimmunity, such as the increased anti-CCP antibody production, observed in this subset.

Among tsDMARDs, Janus kinase (JAK) inhibitors have demonstrated clinical efficacy through the inhibition of the JAK/STAT (signal transducer and activator of transcription) pathway, and, consequently, the signal transduction of pro-inflammatory cytokines such as IL-6 and IFN-γ. Nevertheless, they are associated with a higher risk of serious infections and herpes zoster, necessitating dose adjustments in older adults, for example for tofacitinib, baricitinib, and filgotinib. In patients over 50 years of age with cardiovascular risk factors or a history of smoking, registrational trials have reported increased rates of major adverse cardiovascular events and malignancies, highlighting the need for caution in elderly populations [64,65]. A notable limitation in clinical research is the underrepresentation of older adults in randomized controlled trials, which hampers evidence-based therapeutic decisions in this age group (Table 1).

Beyond infectious risk, elderly RA patients require enhanced oncologic surveillance when treated with advanced therapies, as aging itself is associated with an increased baseline risk of malignancy. RA has been linked to a modestly elevated risk of melanoma, lymphohematopoietic cancers, and, in men, lung, liver, and esophageal cancers [39]. Retrospective cohort studies and reports have highlighted an increased incidence of cervical cancer in women with RA, particularly among those treated with infliximab, including women over the age of 60 [63,64,65,67,68,69,70]. However, beyond this association, a direct causal effect of the drug on this type of malignancy has not been demonstrated to date, nor has it been established for other TNF inhibitors or other cancer types. Similarly, for tocilizumab, abatacept, and rituximab, within the limits of current evidence, no significant increase in overall cancer risk has been observed compared to patients treated with csDMARDs or the general population, with the exception of cutaneous squamous cell carcinoma in patients treated with abatacept [71].

It remains essential to monitor these data longitudinally over extended periods and to carefully assess the risk–benefit profile of biologic therapies in RA and EORA. This includes regular cancer screenings, even in older populations, in particular for cervical cancer—also in women over 60 years—skin cancers, and lymphomas. Indeed, older adults treated with TNF inhibitors appear to have an increased risk of non-melanoma skin cancer overall, non-Hodgkin lymphoma, and particularly follicular lymphoma [72].

## 4. Osteoporosis: A Pathophysiological and Immunological Perspective of Inflammaging

Osteoporosis is a pathological condition characterized by reduced bone mass rarefaction of trabecular bone and thinning of cortical bone, ultimately increasing the risk of fractures. The most common types include postmenopausal and senile osteoporosis. Among the various generalized or regional forms, osteoporosis secondary to joint inflammation is also included, particularly in the context of arthritic processes [88].

At the molecular level, bone resorption can be assessed through specific biomarkers of bone turnover, such as bone-specific alkaline phosphatase (ALP), C-terminal telopeptide of type I collagen (CTX or β-crosslaps), and Procollagen Type 1 N-Terminal Propeptide (P1NP). According to the North American Menopause Society (NAMS) Position Statement, advancing age is a major risk factor for bone loss, with osteoporosis disproportionately affecting women at a female-to-male ratio of approximately 10:1.5 [89,90]. Additional risk factors include low body mass index (BMI), smoking, genetic predisposition, gastrointestinal malabsorption, chronic liver disease, inflammatory bowel disease, and chronic kidney disease, particularly in older adults [91].

From a genetic standpoint, monogenic disorders such as Gaucher disease, hypophosphatasia, juvenile osteoporosis, and osteogenesis imperfecta are implicated [92]. Moreover, in fewer than 30% of cases, polymorphisms in genes encoding type I collagen (e.g., COL1A1), estrogen receptors (ERs), and the vitamin D receptor (VDR)—notably the FokI polymorphism—may structurally alter receptor function, potentially impairing biological signaling [93,94,95]. Screening for anti-tissue transglutaminase IgA antibodies, in the absence of selective IgA deficiency, and/or anti-endomysial antibodies is essential to identify underlying celiac disease, a recognized secondary cause of osteoporosis [96,97]. Similarly, elevated serum tryptase levels may reveal occult mastocytosis [98]. These factors may synergistically contribute to osteoporosis in elderly individuals, both by increasing the likelihood of comorbid conditions and by promoting prolonged exposure to immunoendocrine and metabolic stressors in genetically and epigenetically predisposed individuals, across the lifespan [99,100,101].

There is an extensive cross-talk between the immune and skeletal systems, mediated by a wide array of cytokines, including interleukins IL-1, IL-4, IL-6, IL-10, IL-12, and IL-17, with either pro- or anti-resorptive effects, thus establishing a functional “immuno-skeletal interface” [102]. Adipocytes also contribute significantly to bone homeostasis in aging by secreting adipokines such as leptin and adiponectin, which modulate osteoblasts, osteoclasts, osteoprotegerin, RANKL, 25-hydroxyvitamin D, insulin-like growth factor-binding protein-2 (IGFBP2), and fibroblast growth factor 23 (FGF-23) via direct and neuroimmune pathways, which are also involved in sarcopenia [103,104,105].

In particular, key signaling pathways involved in bone remodeling include the RANK/RANKL/osteoprotegerin (OPG) axis and the Wnt/β-catenin pathway, with sclerostin and Dickkopf-1 (DKK1) acting as potent inhibitors of the latter [106]. Receptor activator of nuclear factor kappa-B ligand (RANKL), a member of the TNF superfamily, is the principal regulator of osteoclast differentiation. It is expressed by osteoblasts and immune cells such as T lymphocytes and APCs [107], which also secrete pro-inflammatory cytokines (e.g., IL-6, IL-17, TNF-α, IFN-γ) that promote osteoclastogenesis and bone loss [108]. RANK, the receptor for RANKL, is expressed on osteoclast precursors and mature osteoclasts.

The RANKL–RANK interaction activates downstream signaling cascades, including nuclear factor kappa-light-chain-enhancer of activated B cells (NF-κB), mitogen-activated protein kinases (MAPKs), and nuclear factor of activated T cells 1 (NFATc1), thereby promoting osteoclast differentiation, activation, and survival. OPG, a soluble decoy receptor secreted by osteoblasts, binds RANKL and inhibits osteoclastogenesis. The RANKL/OPG ratio is a key determinant of the balance between bone resorption and formation [109].

The Wnt/β-catenin pathway supports osteoblast differentiation and bone formation, while also enhancing OPG expression and indirectly suppressing osteoclastogenesis [110]. Wnt glycoproteins bind frizzled and low-density lipoprotein receptor-related protein 5/6 (LRP5/6), stabilizing β-catenin and enabling its nuclear translocation, which activates osteogenic gene transcription [111,112]. Moreover, in this context, recent studies suggest a potential regulatory role for the IL-33/IL-31 axis in osteoporosis [113].

IL-31 promotes osteoclastogenesis both directly, by inducing the differentiation of osteoclast precursors, and indirectly, by stimulating monocytes, APCs, and Th1/Th17 lymphocytes to release metalloproteinases, chemokines, and inflammatory cytokines. This pro-inflammatory loop exacerbates bone loss, supporting a role for IL-31 in postmenopausal and senile osteoporosis, both regarded as age-related inflammatory conditions [114]. IL-31 is a pro-inflammatory cytokine belonging to the gp130/IL-6 cytokine family and is primarily secreted by activated CD45RO^+^ Th2 cells. Although it is best known for its role in skin diseases [115] such as atopic dermatitis, chronic pruritus [116]—a common condition in the elderly [117]—and cutaneous lymphomas, elevated serum IL-31 levels have also been reported in postmenopausal women with low bone mineral density (BMD). While IL-31 levels do not appear to directly correlate with fracture severity, they are positively associated with age, suggesting a potential link between IL-31, aging, and bone inflammation, acting as a mirror of bone-specific immunosenescence and inflammaging.

Conversely, IL-33, a Th2-type cytokine produced by stromal cells in response to inflammatory stimuli, exerts a protective role in bone homeostasis [118]. IL-33 inhibits osteoclastogenesis by blocking the differentiation of myeloid precursors and simultaneously promotes the differentiation and maturation of osteoblasts from mesenchymal stem cells. Although IL-33 stimulates IL-31 production by Th2 cells, it also suppresses RANKL-induced osteoclast formation, resulting in a net anti-resorptive effect.

The role of the IL-33/IL-31 axis in osteoporosis is multifactorial, influenced by reciprocal cytokine regulation, aging, sex steroid deficiency, and interaction with other cytokines such as IL-4, IL-10, IL-17, and transforming growth factor β1 (TGF-β1) [119]. Regulatory T cells (Tregs), for instance, exert an osteoprotective function by producing IL-4, IL-10, and TGF-β1, which inhibit both Th17 cells and osteoclast differentiation. These observations support the concept that postmenopausal and senile osteoporosis is not merely a degenerative condition but rather an immune-mediated disease: IL-31 and IL-33 act as central regulators within the broader cytokine network, along with the delicate balance between pro- and anti-osteoclastogenic/osteogenic factors in the RANK/RANKL/OPG and Wnt/β-catenin pathways, negatively modulated by sclerostin and DKK1 [120,121,122].

On the other hand, sclerostin is a 24 kDa monomeric glycoprotein encoded by the *SOST* gene and is a key negative regulator of bone formation, highly expressed by osteocytes [69]. It binds LRP5/6 receptors and prevents their interaction with Wnt ligands, thereby inhibiting β-catenin signaling and suppressing osteogenesis. Sclerostin expression increases under conditions of mechanical unloading, estrogen deficiency, and aging. Its inhibition has been shown to exert anabolic effects on bone.

DKK1 is another soluble Wnt antagonist secreted by osteocytes, osteoblasts, immune cells, and neoplastic cells. It binds to LRP5/6 and Kremen co-receptors, leading to their internalization and degradation, thereby persistently inhibiting the Wnt pathway. Unlike sclerostin, DKK1 is more broadly expressed in inflammatory and malignant settings [123,124].

Estrogens suppress the expression of RANKL, sclerostin, and DKK1, thereby exerting a protective role on bone homeostasis. Conversely, estrogen deficiency and increased levels of pro-inflammatory cytokines, such as TNF-α, IL-1β, IL-6, and IL-17, upregulate RANKL, DKK1, and sclerostin expression, shifting the balance toward bone resorption (Figure 2) [125].

Rheumatoid arthritis (RA) is characterized by synovial inflammation driven by pro-inflammatory cytokines and autoantibody-mediated mechanisms in seropositive forms. In osteoporosis, the anti-osteoporotic Wnt/β-catenin pathway is inhibited by DKK1 and sclerostin. Among shared mechanisms, IL-6, a pivotal cytokine in RA pathophysiology and progressively increased during pathological aging, enhances bone resorption, supporting the immune-mediated inflammatory features of osteoporosis. Increased RANKL and decreased osteoprotegerin (OPG) contribute to bone erosion and damage in RA and to bone fragility and fracture risk in osteoporosis, respectively. Within the IL-33/IL-31 axis, IL-33 is a Th2-type inflammatory cytokine with a bidirectional role in osteoporosis. These shared mechanisms, amplified during inflammaging, make RA and osteoporosis two prototypes of immunosenescence.

### Molecular Targeted Therapy of Osteoporosis

The molecular pathways involved in bone remodeling also serve as key therapeutic targets for two biologic agents that play a crucial role in the prevention and treatment of osteoporosis, alongside conventional antiresorptive drugs such as bisphosphonates and the osteoanabolic agent teriparatide [126,127].

Denosumab, a fully human IgG2 monoclonal antibody against RANKL, is indicated for both primary and secondary prevention of osteoporosis in postmenopausal women and in men over 50 years of age who are at increased risk of fracture or undergoing adjuvant hormonal therapy. By inhibiting the formation, activity, and survival of osteoclasts, denosumab reduces bone resorption in both cortical and trabecular bone compartments [73].

Beyond skeletal effects, the RANK/RANKL pathway is also implicated in extra-skeletal processes, particularly in the immune system. RANKL is involved in thymic development and T-cell selection, as well as in the organogenesis of secondary lymphoid tissues. It is also critical for the differentiation of microfold cells in Peyer’s patches, which are specialized in translocating luminal antigens for antigen presentation. Moreover, it plays a pivotal role in dendritic cell survival and dendritic cell–T cell interactions, ultimately regulating central T-cell tolerance, peripheral T-cell function and survival, and efficient antigen presentation [74,75].

With appropriate monitoring, denosumab can be safely administered even in older adults [76,77] with an inherently increased risk of infection. In the FREEDOM trial, a multicenter, randomized, double-blind, placebo-controlled study, a slight increase in serious infections affecting the gastrointestinal, urogenital, and otologic systems, as well as in cases of endocarditis, was observed in the denosumab group compared to placebo. However, no consistent correlation with treatment duration or cumulative exposure was established [78]. Moreover, long-term data support the efficacy and safety of denosumab for up to 10 years of continuous use [79,80,81].

Romosozumab is another monoclonal antibody approved for the treatment of severe osteoporosis in postmenopausal women at high risk of fracture [85]. It exerts a dual mechanism of action—antiresorptive and osteoanabolic—which distinguishes it from other available therapies and has earned it the designation of a “bone builder”. Romosozumab targets sclerostin, a negative regulator of the Wnt/β-catenin signaling pathway, thereby promoting bone formation and reducing resorption. Its approval is primarily based on pivotal trials conducted in women, which demonstrated significant reductions in fracture risk [86]. Although romosozumab has been shown to increase BMD in men, robust data from randomized controlled trials in the male population are lacking, and the drug has not yet received Food and Drug Administration (FDA) approval for use in men [87].

Notably, the ARCH trial revealed a higher incidence of serious cardiovascular adverse events (2.5%) in the romosozumab group during the double-blind phase, compared to 1% in the alendronate group. These events included ischemic heart disease, cerebrovascular events, and heart failure. However, this increased risk was not confirmed in the primary analysis, which included both the double-blind and open-label phases [128].

With regard to the pharmacological modulation of additional molecular targets, although molecules directed against IL-31 and IL-33 are currently in preclinical development or clinical approval for atopic dermatitis/prurigo nodularis [129,130] and asthma/COPD [131,132], respectively, to our knowledge, no monoclonal antibodies or small molecules targeting these interleukins or their receptors are currently under investigation for osteoporosis. this represents an interesting avenue for the exploration of potential innovative therapies in this field.

## 5. Gender Considerations

Immunosenescence is closely linked to the hormonal regulation of the immune system, particularly involving sex hormones such as estrogens, progesterone, and androgens. Immunosenescence can be conceptualized as a negative control model, in contrast to the hormone-driven positive regulation that influences sex-dependent susceptibility to osteoimmunological diseases. While these conditions predominantly affect women during their reproductive years, gender differences tend to diminish with advancing age. In certain diseases, such as RA, the male-to-female ratio appears more balanced, tending toward 1:1, whereas a slight male predominance may even emerge in late-onset forms. The primary explanation for this phenomenon is the age-specific hormonal shift, which places the condition of EORA in the postmenopausal period or generally after the age of 50–60. In particular, in men, it is linked to a reduction in the immunosuppressive contribution of testosterone, creating a state of increased susceptibility to immune-mediated processes during this stage of life [133].

Osteoporosis is traditionally considered as a female-predominant condition, and romosozumab serves as an interesting example of gender-specific pharmacology; however, it also exhibits distinct male-specific characteristics. Fragility fractures pose a major public health concern due to their association with increased mortality, loss of independence, and substantial healthcare costs [134,135]. For example, hip fractures are associated with approximately 5% mortality within one month and 20% within one year [136,137,138]. Men account for nearly 40% of all osteoporotic fractures, with incidence rates steadily rising. However, due to persistent gender disparities in clinical management, only about 10% of men receive appropriate preventive treatment, compared to significantly higher rates in women. Notably, more than 50% of osteoporosis cases in men are classified as secondary [139].

In contrast to postmenopausal osteoporosis in women, the most common etiologies in men include hypogonadism and chronic glucocorticoid therapy [140]. According to Endocrine Society guidelines, testosterone replacement therapy is recommended in men with confirmed hypogonadism, when testosterone levels are below 200 ng/dL on two separate measurements, in absence of any contraindications. Although not a first-line option for fracture prevention, testosterone therapy may be considered in patients who are ineligible for standard osteoporosis treatments, owing to its ability to improve BMD and reduce bone turnover markers [141]. Nevertheless, current evidence does not support its efficacy in reducing fracture risk, and testosterone is not indicated in eugonadal individuals. In older men with normal testosterone levels, no clinically significant improvements in bone mineral density have been demonstrated [142].

Alcohol consumption—more prevalent among men—is an additional gender-related risk factor for osteoporosis [143]. Secondary causes should also be considered, including celiac disease, which is more common in women [144], and systemic mastocytosis, which shows a stronger association with male osteoporosis [145].

In conclusion, in men, osteoporosis is much more often secondary compared to women and is more strongly influenced by chronic diseases and the use of osteopenic drugs. Therefore, although bone resorption in men tends to be slower and more gradual and osteoporosis is diagnosed later, susceptibility to complications and the risk of post-fracture mortality are higher. In particular, unlike the impact of estrogen loss on vertebral trabecular bone in women, in men, the loss of cortical bone is relatively greater, begins in midlife, and accelerates after the age of 70, thereby increasing the risk of non-vertebral fractures. Nevertheless, it should also be considered that men, compared to women, have larger bones and thicker cortices, which overall are bigger and stronger [140].

## 6. Discussion

We reviewed RA and osteoporosis, focusing on their molecular, clinical, and therapeutic features in the elderly, through the integrated framework of immunosenescence and inflammaging. Furthermore, we highlighted how these two seemingly distinct conditions share several points of intersection. In postmenopausal women and during immunosenescence, levels of estrogen and TGF-β, which support bone formation, decline, while pro-resorptive cytokines such as IL-1, IL-6, and TNF-α increase. This shift is the result of immuno-endocrine interactions associated with aging. 

Such changes in the cytokine environment stem from the functional alterations of senescent T cells and lead to a chronic, low-grade inflammatory state at the bone interface, thus positioning osteoporosis as an inflammatory, immune-mediated disease [146], as well as RA (Figure 3).

Regarding age-related rheumatologic inflammatory processes and their relationship with secondary osteoporosis, an interesting intersection emerges between the molecular pathways of bone metabolism and osteoarticular damage. The RANK/RANKL signaling pathway is central, not only to osteoclastogenesis but also to bone erosion in RA. Similarly, DKK1 has been implicated in ankylosing spondylitis and pathological new bone formation, further highlighting the interplay between bone remodeling and immune dysregulation. It is therefore unsurprising that this antiresorptive mechanism may, from an osteoimmunological perspective, exert a protective role in preventing the bone loss typical of the erosive processes of RA. Preliminary evidence supports the potential inhibition of joint damage progression when denosumab is administered in combination with csDMARDs [82].

This intersection extends into the field of osteoimmuno-oncology; in fact, denosumab has demonstrated efficacy not only in osteoporosis but also in the treatment of bone metastases from solid tumors, multiple myeloma [83],and in breast cancer patients receiving aromatase inhibitors in the adjuvant setting [84].

However, the systemic role of the RANK/RANKL pathway, particularly in immune tissues, warrants caution. In elderly patients, especially those with major infections, sepsis, or undergoing immunosuppressive or oncologic treatments, denosumab may exacerbate susceptibility to infection. Similar concerns apply to advanced biologic therapies for RA in older adults, as well as anti-IL-6, anti-CD20 agents, and JAK inhibitors [147].

In EORA, a treat-to-target strategy may be appropriate for selected patients over 60 years with low comorbidity burden, preserved cognition, and minimal polypharmacy. In frailer patients, a personalized treatment plan is essential, considering cardiovascular, infectious, hepatic, renal, and oncologic risks. Corticosteroid use should be minimized, and preventative strategies, as well as vaccinations, lipid control, and lifestyle interventions, should be prioritized [66]. Drug dosing must be carefully tailored based on renal and hepatic function. For example, methotrexate should be started at reduced doses [148,149], and biologics with shorter half-lives (e.g., etanercept, abatacept) may be preferable. Moreover, polypharmacy requires meticulous attention to drug–drug interactions.

One of the key limitations in geriatric pharmacology is the elevated risk associated with systemic comorbidities compounded by aging. The underrepresentation of older adults in clinical trials contributes to a gap in evidence-based management for this population. Conversely, adverse events reported in this age group may reflect baseline geriatric syndromes rather than direct drug toxicity. Nevertheless, biological drugs for osteoporosis must necessarily be evaluated with a focus on immunosenescence, as their use is inherently directed toward older populations. In the case of denosumab—unlike biological drugs used for arthritis—the average age of patients, in real-world studies and treatment persistence analyses, ranges between 70 and 79 years, with 30–50% of patients being older than 80 [76,77,78,79,80].

## 7. Conclusions

Immunosenescence and “inflammaging” are central to shaping the immune and endocrine landscapes in older adults, profoundly impacting bone health and the development of osteoimmunological pathologies. Age-related hormonal changes and the variations in cytokine imbalance, such as IL-2, IL-6, TNF-α levels, and IL-33/IL-31 axis, as well as serum antibody concentrations and the upregulation of RANKL, DKK1, and sclerostin, drive a chronic, low-grade inflammatory state that underpins many age-related diseases, as well as RA and osteoporosis, both with features of inflammatory immuno-mediated diseases.

Personalized therapeutic strategies should be prioritized over a strict treat-to-target approach in elderly patients, particularly in the context of polypharmacy, multiple comorbidities, and heightened infectious, oncologic, and cardiovascular risks. The goal is to balance safety and efficacy, favoring the lowest effective doses and therapies with shorter half-lives when possible. Targeted therapies represent a valuable addition to the therapeutic arsenal in managing bone immunosenescence, at times overcoming the limitations of conventional treatments and helping to optimize the prevention and treatment of age-specific complications.

Even though abatacept demonstrates an overall favorable safety profile, it is important to account for the possible increased risk of certain malignancies associated in particular with TNF inhibitors, such as non-melanoma skin cancer and non-Hodgkin lymphoma in older adults, and cervical cancer also in women over 60, as well as cutaneous squamous cell carcinoma linked to abatacept itself. In addition, infectious risks should be considered with tocilizumab, rituximab, and JAK inhibitors. The latter, together with romosozumab, also carry cardiovascular warnings that may require caution or dose adjustments in elderly patients.

On the other hand, by acting on their specific molecular signaling pathways, denosumab—with its potential role in limiting bone erosions and preventing skeletal metastases—and romosozumab—with its bone builder action—are monoclonal antibodies, potentially able to modulate inflammaging, within the context of osteoimmunology.

Further research is needed to better characterize the molecular pathways underlying immunosenescence and to develop translational, targeted therapies [150] aimed at modulating inflammaging. Special attention should be given to the elderly population, ensuring their appropriate representation in clinical trials and addressing their specific management needs.

## Figures and Tables

**Figure 1 ijms-26-09268-f001:**
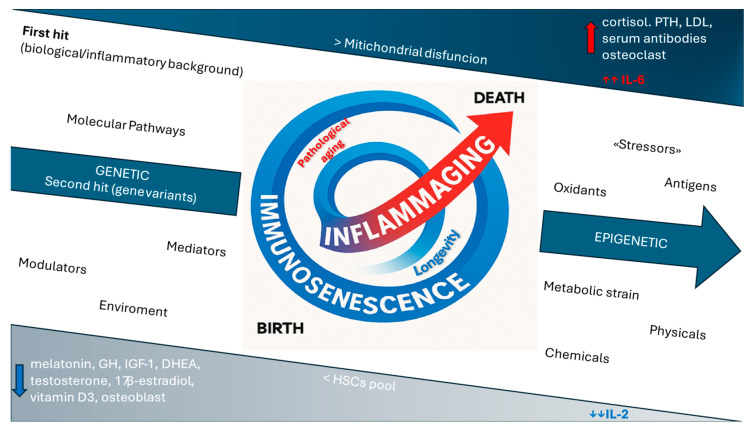
The integrated model of immunosenescence and inflammaging.

**Figure 2 ijms-26-09268-f002:**
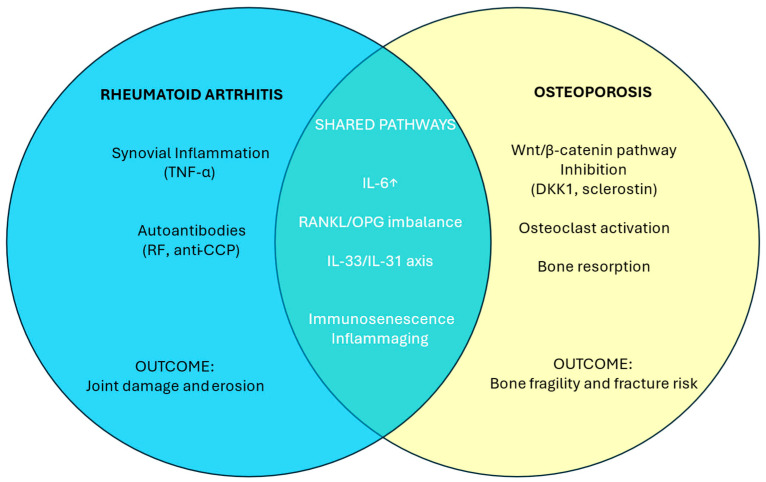
Side-by-side representation of rheumatoid arthritis and osteoporosis pathways: shared mechanisms and distinct features.

**Figure 3 ijms-26-09268-f003:**
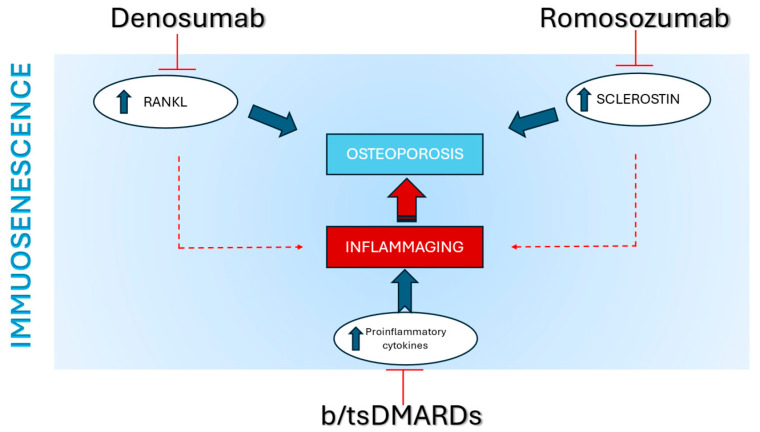
Osteoporosis and inflammaging and their modulation through targeted therapies in immunosenescence. Immunosenescence leads to an increased production of proinflammatory cytokines and elevated expression of RANKL and sclerostin, resulting in a shift toward bone resorption. The monoclonal antibodies denosumab and romosozumab, by targeting RANKL and sclerostin respectively, modulate the RANK/RANKL and Wnt/β-catenin pathways in an anti-resorptive direction. The increase in proinflammatory cytokines contributes to chronic inflammation at the immuno-skeletal interface and inflammaging, similarly to that observed in immune-mediated arthritic conditions. Biological therapies for osteoporosis, along with biologic disease-modifying antirheumatic drugs (bDMARDs) and targeted synthetic DMARDs (tsDMARDs), collectively modulate the chronic low-grade inflammation (inflammaging) associated with osteoporosis and osteoimmunological diseases in immunosenescence.

**Table 1 ijms-26-09268-t001:** Target therapies in osteoimmunology and their management in immunosenescence from real-world data and clinical trials.

Drug	Molecular Target	Immunosenescence Management
Etanercept	TNF-α (Inflammatory cytokine)	Shorter half-life [66].
Infliximab	TNF-α (Inflammatory cytokine)	Increased risk of cervical cancer (screening) [65,67,68,69,70].
Other TNFi	TNF-α (Inflammatory cytokine)	Increased risk of non-melanoma skin cancer and non-Hodgkin lymphoma in older adults (screening) [71,72].
Tocilizumab	IL-6 (Inflammatory cytokine)	Higher risk of infection [59,71].
Rituximab	CD 20 (B cell surface protein)	Higher risk of infection [60,61].
Abatacept	CD80/CD86 (CTLA-4 immunomodulatory pathway)	Favorable safety and tolerability profile; shorter half-life; increased risk of cutaneous squamous cell carcinoma (screening) [62,71].
Baricitinib Tofacitinib Fingotinib	JAK (JAK/STAT pathway; inflammatory signals transduction)	Considers cardiovascular risk factors, Herpes Zoster infection, and dose adjustments [64,65].
Denosumab	RANKL (Proinflammatory and osteoclastogenic cytokine)	Antiresorptive treatment in OP; reduction of bone erosions progression in RA and tumor metastasis risk; considers infectious risk factors, hypocalcemia, and renal function [73,74,75,76,77,78,79,80,81,82,83,84].
Romosozumab	Sclerostin (Bone formation inhibiting protein)	Bone builder; considers cardiovascular risk factors [85,86,87]

TNF-α: tumor necrosis factor α; TNFi: tumor necrosis factor inhibitor; IL-6: interleukin 6; CD: cluster of differentiation; CTLA-4: cytotoxic T-lymphocyte antigen 4; JAK: Janus kinase; STAT: signal transducer and activator of transcription; RANKL: receptor activator of nuclear factor kappa-B ligand; OP: osteoporosis; RA: rheumatoid arthritis.

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
