# Peer review of "Rheumatoid Arthritis and Osteoporosis as Prototypes of Immunosenescence in Osteoimmunology: Molecular Pathways of Inflammaging and Targeted Therapies"

_ijms, 2025, doi:10.3390/ijms26199268_

Round 1

Reviewer 1 Report

Comments and Suggestions for Authors

This paper examines the effects of aging on immune and inflammatory function, focusing on the clinical characteristics, molecular mechanisms, and corresponding treatments of rheumatoid arthritis and osteoporosis caused by these alterations. It also offers valuable recommendations for medication use in treating diseases in the elderly and provides new research avenues for future studies of geriatric diseases.

But there are still have some issues need to modify:

  1. Figure1: Does these factors ( “genetic predisposition, epigenetic regulation, and molecular and metabolic pathways, lifelong environmental interactions and exposure to antigenic, oxidative, metabolic, chemical, and physical stressor” ) has the time course from birth to death? Or it randoms? These factors with this way to show looks strange
  2. Table1: references should be added in each drug treatment.
  3. References 89 is not a good place. Need to check this.

Author Response

1.Figure1: Does these factors ( “genetic predisposition, epigenetic regulation, and molecular and metabolic pathways, lifelong environmental interactions and exposure to antigenic, oxidative, metabolic, chemical, and physical stressor” ) has the time course from birth to death? Or it randoms? These factors with this way to show looks strange.

Dear Reviewer,

First of all, thank you for grasping the essence of this paper and for your valuable suggestions to enhance the overall quality of the work. In particular, we appreciate your encouragement to strengthen Figure 1, which is central to representing the concepts of immunosenescence and inflammaging, foundational for this review. Indeed, we agree that at first glance, the arrangement of the mentioned factors might appear disordered and random.

In this revised version, we have aimed to organize the factors into those fixed from birth and those dynamic and cumulative throughout life. Specifically, in the white area, the listed factors refer to the biological and inflammatory background that constitutes the “first hit” in inflammaging; the new arrow, crossing the entire figure, represents genetics and its variants (second hit) as well as epigenetics, thereby illustrating the second hit theory graphically.

The outcomes of longevity versus pathological aging have been repositioned close to the spiral of immunosenescence and inflammaging. The triangles depicting the immuno-endocrine alterations of immunosenescence have been enriched with the cellular component of mitochondrial dysfunction and the hematopoietic stem cell pool, completing the definition of immunosenescence.

With this graphical revision, we hope to have made the figure more intuitive, comprehensible, and orderly, improving its overall clarity and quality.

  1. Table1: references should be added in each drug treatment.

Thank you for pointing out the need to include references to the table. We have added the bibliography for each drug in relation to the aspects highlighted, as suggested. In this way, we hope to have made the table more readable, accessible, and scientifically rigorous.

  1. References 89 is not a good place. Need to check this.

Dear Reviewer,

Thank you for pointing out the placement of reference 89. We have reviewed it and, following your suggestion, we have moved it to line 380 in relation to “chronic pruritus,” since the cited work pertains to the influence of age on skin conditions, dermatitis, and chronic pruritus (paragraph 8 of that paper). We hope that with this adjustment we have found a more appropriate placement for this reference and that it aligns with your expectations.

Reviewer 2 Report

Comments and Suggestions for Authors

The paper "Rheumatoid arthritis and osteoporosis as prototypes of immunosenescence in osteoimmunology: molecular pathways of inflammaging and targeted therapies” by Aitella et al. has a scientific value that can be considered high, as it demonstrates significant mechanistic depth and successfully synthesizes the fields of osteoimmunology and geriatric rheumatology. However, its clinical applicability is moderate; while there is a strong awareness of the associated risks, there are fewer actionable guidelines available for practitioners. In terms of novelty, the integration presented is valuable but not entirely new, highlighting an area of ongoing research rather than groundbreaking findings. Additionally, the readability of the content may pose challenges for non-specialists due to its dense technical nature, making it less accessible to a broader audience. The paper is considered acceptable for publication, pending minor revisions related to the following points:

  1. Introduction, lines 28–50: The review does not state how articles were selected or whether a systematic search was done. Without methodology, it is unclear whether the review is comprehensive or selective, which weakens scientific rigor.
  2. Given the dense mechanistic content, an extra figure could help readers, especially clinicians or non-specialists, navigate the complexity. For example, I suggest including additional illustrations on comparative disease pathways, particularly a side-by-side pathway map of RA vs. osteoporosis, showing where they diverge and where they overlap (IL-6, RANKL/OPG, DKK1, IL-33/IL-31). I think this would visually emphasize the review’s main argument that both are prototypes of immunosenescence.
  3. Line 225-262 and 477 and so on: Some therapies (e.g., abatacept, denosumab) are well discussed, while others (e.g., JAK inhibitors, conventional DMARDs) receive less mechanistic or comparative detail. Please provide additional discussion on this in the revised manuscript.
  4. Line 440-474: Although a “Gender Considerations” section exists, male osteoporosis and late-onset RA are only briefly addressed relative to female-focused data. This paragraph should be improved in the revised manuscript.

Author Response

  1. Introduction, lines 28–50: The review does not state how articles were selected or whether a systematic search was done. Without methodology, it is unclear whether the review is comprehensive or selective, which weakens scientific rigor.

Dear Reviewer,
first of all, thank you very much for your kind appreciation of our work and for the valuable comments and suggestions provided. We are especially grateful for pointing out this methodological gap, which we believe we have addressed according to your indications at line 43, by adding the research methodology, keywords, and the selective focus of the work. We hope that this has strengthened the overall scientific rigor of the manuscript.

  1. Given the dense mechanistic content, an extra figure could help readers, especially clinicians or non-specialists, navigate the complexity. For example, I suggest including additional illustrations on comparative disease pathways, particularly a side-by-side pathway map of RA vs. osteoporosis, showing where they diverge and where they overlap (IL-6, RANKL/OPG, DKK1, IL-33/IL-31). I think this would visually emphasize the review’s main argument that both are prototypes of immunosenescence.

Thank you for the observation and for the excellent idea! Following your suggestion, we created this side-by-side representation of the main molecular pathways in rheumatoid arthritis and osteoporosis, summarizing the key pathophysiological steps and their outcomes. In addition, with the support of an explanatory caption, we aimed to highlight the shared pathway and the main key points to facilitate understanding of the discussion and the overall narrative of the review, even for non-specialist readers. We hope this meets your expectations.

            3.Line 225-262 and 477 and so on: Some therapies (e.g., abatacept, denosumab) are well discussed, while others (e.g., JAK inhibitors, conventional DMARDs) receive less mechanistic or comparative detail. Please provide additional discussion on this in the revised manuscript.

Dear Reviewer, thank you for pointing out this critical issue, which could have made comprehension difficult for a non-specialist reader. As suggested, we have addressed this particularly in paragraph 3.1, by integrating the main mechanisms of action of the drugs where not previously mentioned  (for example cDMARDs and JAK inhibitors suggested) and, where possible, by providing some clarifying details for those already described (for example lines 449, 553), in order to facilitate a clearer understanding of the mechanisms related to inflammation.
 Also table 1 has improved in that sense.”

          4.Line 440-474: Although a “Gender Considerations” section exists, male osteoporosis and late-onset RA are only briefly addressed relative to female-focused data. This paragraph should be improved in the revised manuscript

Dear Reviewer,

We recognized the need to standardize the discussion on gender; therefore, we thank you for your comment, which allowed us to also improve the aspects related to gender medicine. Accordingly, we have made the appropriate additions to the paragraph at lines 489 and 520.

Reviewer 3 Report

Comments and Suggestions for Authors

This review focuses on the roles of immunosenescence and chronic low-grade inflammation (inflammaging) in the pathogenesis of rheumatoid arthritis (RA) and osteoporosis. It highlights how age-related endocrine immune alterations, including reduced sex hormone levels, cytokine imbalance (e.g., decreased IL-2 and increased IL-6), and the upregulation of RANKL, DKK1, and sclerostin, collectively drive dysfunction of the skeletal and immune systems in the elderly. The article emphasizes the shared molecular mechanisms between RA and osteoporosis within the framework of osteoimmunology and discusses the features and risks of both conventional and targeted therapies (including methotrexate, biologics, JAK inhibitors, denosumab, and romosozumab) in older patients. However, several areas require improvement or clarification, as outlined below:

  1. In the abstract, the sentence “In the elderly, the decline in sex hormones and IL-2, along with changes in TNF-α, pro-inflammatory cytokines and the IL-33/IL-31 axis, the increase in IL-6 and serum autoantibodies, and the upregulation of RANKL, DKK1, and sclerostin…” lists multiple items in a disorganized manner; restructuring is recommended for clarity.
  2. The review confines “immunosenescence” to immune endocrine alterations, whereas recent literature also emphasizes hematopoietic stem cell exhaustion and mitochondrial dysfunction. Should this definition be broadened?
  3. In Section 2, the discussion on inflammaging is brief; additional content would strengthen the manuscript.
  4. Both IL-6 and IL-2 exert bidirectional effects in immune regulation, yet Figure 1 depicts only a unidirectional impact. Does this accurately reflect current knowledge?
  5. The underlying mechanisms of the “second hit” model in Figure 1 are not fully explained and should be elaborated.
  6. The molecular targets in Table 1 should be closely related to inflation. It is recommended to supplement and modify them.
  7. Section 2.1 (“Implications in Gerontorheumatology”) is tangential to the main topic and may need to be deleted or integrated into related content.
  8. The table discusses targeted therapies in osteoimmunology and their roles in immunosenescence but lacks supporting references. What evidence underpins the safety risk statements for agents such as abatacept, rituximab, and JAK inhibitors, real-world data or clinical trials?
  9. The manuscript advises lowering the starting dose of methotrexate but provides no specific recommendations or guideline citations. Should international guidelines (e.g., EULAR, ACR) be referenced?
  10. Many arguments link IL-6 elevation and IL-2 reduction caused by immunosenescence to RA and osteoporosis. Are there direct clinical or animal model studies validating these mechanisms?
  11. While IL-33 is suggested as pivotal in osteoporosis and immunosenescence, the review does not address whether antagonists or agonists are in clinical development. Should the feasibility of this pathway as a treatment target be evaluated?

Author Response

  1. In the abstract, the sentence “In the elderly, the decline in sex hormones and IL-2, along with changes in TNF-α, pro-inflammatory cytokines and the IL-33/IL-31 axis, the increase in IL-6 and serum autoantibodies, and the upregulation of RANKL, DKK1, and sclerostin…” multiple items in a disorganized manner; restructuring is recommended for clarity.

Dear Reviewer,

First of all, we would like to sincerely thank you for your careful evaluation of this review and for the numerous suggestions, which have undoubtedly helped us improve the overall quality of the paper at various levels. We also appreciate your observation regarding a certain degree of disorganization of the molecular elements listed in the abstract, which was the result of our attempt to summarize and schematize them.

We have therefore restructured the sentence in question by dividing the key elements into three categories: 1) immunoendocrine alterations; 2) inflammatory cytokines; and 3) osteoporosis-related markers. We hope that this revision meets your request and that the abstract is now clearer and more effective overall.

2. The review confines “immunosenescence” to immune endocrine alterations, whereas recent literature also emphasizes hematopoietic stem cell exhaustion and mitochondrial dysfunction. Should this definition be broadened?

Dear Reviewer,

Thank you very much for highlighting the limitations of the definition of immunosenescence. Although the focus of our work was on the immuno-endocrine alterations underlying this condition, we agree that incorporating the concepts of mitochondrial dysfunction and hematopoietic stem cell exhaustion can further enhance the scientific rigor of the paragraph in question. Therefore, we have included these aspects to complement the cellular and molecular alterations already discussed. This addition also strengthens the characterization of senescence from a properly cellular perspective.

3. In Section 2, the discussion on inflammaging is brief; additional content would strengthen the manuscript.

Dear Reviewer, thank you for pointing out the discrepancy between the two parts of the section. As per your suggestion, we have expanded the part concerning inflammaging by including cellular and molecular references and a link to metaflammation and macroph-aging. We hope that this integration meets your expectations and has contributed to improving the overall quality of the manuscript.

4. Both IL-6 and IL-2 exert bidirectional effects in immune regulation, yet Figure 1 depicts only a unidirectional impact. Does this accurately reflect current knowledge?

Dear Reviewer, thank you very much for raising this interesting and important point of high immunological relevance. For the sake of simplicity, we mistakenly provided an overly simplistic and unidirectional view of the actions of IL-2 and IL-6. We therefore considered it appropriate, even though not directly related to the figure, to make a specific reference to their bidirectionality in the text (line 99) with the relative references. In the figure itself, however, for graphical simplification, we specified in the caption that it refers to the pathological aging model. Nonetheless, to convey the idea of a concentration gradient, we have added a double arrow corresponding to the quantitative changes of the two interleukins represented. We hope that, in this way, together with the support of the text and the caption, we have sufficiently introduced the concept of bidirectionality and, at least in part, contextualized the limitations of the graphical interpretation.

5- The underlying mechanisms of the “second hit” model in Figure 1 are not fully explained and should be elaborated.

Dear Reviewer,

Thank you for pointing out that the “second hit” theory, although mentioned in both the text and the figure legend, was not visually represented. Beyond being helpful for the reader, your remark also allowed us to bring greater order and clarity to the figure by dividing the various factors into the first hit (biological and inflammatory background) and the second hit (genetics and its variants, represented by the new transversal arrow).

In particular, this also enabled us to distinguish graphically between factors fixed from birth and those dynamic and cumulative over the course of life, while the outcomes of longevity and pathological aging have been placed within the context of the immunosenescence and inflammaging spiral

6. The molecular targets in Table 1 should be closely related to inflation. It is recommended to supplement and modify them.

Thank you for your comment. The molecular targets highlighted in the table have significant implications from the perspective of osteoimmunology, bone pathophysiology, and inflammation. Therefore, as you rightly suggested, we have enriched Table 1 with the mechanistic connections most closely related to the indicated molecular targets. We hope that this makes  more comprehensible for the reader the table and the direct or indirect connection with inflammation, without excessively burdening it in terms of content or graphics.

7. Section 2.1 (“Implications in Gerontorheumatology”) is tangential to the main topic and may need to be deleted or integrated into related content.

Thank you for the comment. Indeed, structuring a paragraph dedicated to gerontorheumatology diverges from the main focus of the work and would have weighed down the overall structure. However, the references provided represent a useful support for introducing the epidemiological and therapeutic aspects of the elderly in this study. Therefore, we have removed the paragraph as requested and transferred its content to the section “Conventional and Advanced Therapies in Rheumatoid Arthritis: Considerations for the Elderly.

8. The table discusses targeted therapies in osteoimmunology and their roles in immunosenescence but lacks supporting references. What evidence underpins the safety risk statements for agents such as abatacept, rituximab, and JAK inhibitors, real-world data or clinical trials?

Thank you for pointing out the lack of references. We apologize for this omission and have added the relevant references for each drug regarding the highlighted aspects. Since these include both real-world data and clinical trials, we have also specified this in the table title. We hope that, thanks to your suggestion, the table is now overall more accessible and scientifically rigorous

9. The manuscript advises lowering the starting dose of methotrexate but provides no specific recommendations or guideline citations. Should international guidelines (e.g., EULAR, ACR) be referenced?

Thank you for highlighting this interesting point. We apologize for not citing supporting guidelines; however, to our knowledge, there are currently no official recommendations from major scientific societies specifically addressing the management of this issue in the elderly. In fact, it is precisely these gaps, which are encountered daily in clinical practice, that motivated the need to explore certain therapeutic and management aspects for older patients in this work. Nevertheless, to provide adequate bibliographic support, we have added references 148 and 149, including a real-world clinical study. These are added to the previously cited references 56 and 57 from the earlier version of the manuscript regarding this theme. We hope that, in this way, we have effectively and coherently addressed this weakness of the work.

10. Many arguments link IL-6 elevation and IL-2 reduction caused by immunosenescence to RA and osteoporosis. Are there direct clinical or animal model studies validating these mechanisms?

Dear Reviewer, thank you for this comment, which has certainly contributed to enhancing the scientific rigor of the manuscript and the overall quality of the work. In line with the considerations above regarding IL-2 and IL-6 and their discussion in the text, we have specified (line 102) the existence of supporting animal models along with the corresponding bibliographic references.

11. While IL-33 is suggested as pivotal in osteoporosis and immunosenescence, the review does not address whether antagonists or agonists are in clinical development. Should the feasibility of this pathway as a treatment target be evaluated?

Dear Reviewer, thank you for the insightful comment, which allowed us to expand and update the present review with a focus on future perspectives. At the end of the section Molecular Targeted Therapy of Osteoporosis, we have added the current state of knowledge regarding the pharmacological modulation of the IL-33/IL-31 axis and in osteoporosis, along with the relevant references.